# Stability of Extended Earth Berm for High Landfill

**Haoran Jiang [1],\*, Xiaowen Zhou [1] and Ziwei Xiao [2]**

[1]    School of Civil Engineering and Transportation, South China University of Technology, Wushan Rd,
      Guangzhou 510641, China; xwzhou@scut.edu.cn
[2]    China Railway Southern Investment Co., Ltd., Shenzhen 518054, China; dic2006@126.com
\*    Correspondence: 201410101152@mail.scut.edu.cn

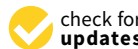

**Featured Application: The results of this paper are suitable for the construction of landfill expansion, especially for high-steep slope engineering due to limited construction space. The design scheme and analysis method in this paper have significance for the safety assessment and management of landfill sites.**

**Abstract:** This study presents a stability analysis of an extended berm reinforced by geotextiles, with a steep slope of 1V:1.1H (vertical: horizontal). Finite element (FE) analyses were carried out to explore the failure mechanism and factor of safety (FOS) of the berm, on which the effect of the strength of geotextiles, leachate level, and anti-slide pile arrangement located at the toe of the berm were considered. It was found that: (1) failure surfaces developed along the interface between the new and the existing berms; (2) the FOS decreased as the leachate increased, and an FOS value of 1.42 could be obtained if the leachate level was controlled at a height of 20 m; (3) the tensile force of geotextiles was far lower than the available strength, which suggested that the geotextile had enough of a safety reserve; and (4) one row of longer piles at the toe of the berm performed better than two rows of shorter piles if the total length of piles was the same. The design and analysis of this project can be used as a reference for landfill expansion. Especially for a site condition with limited space, a geosynthetic-reinforced soil (GRS) berm is a safe, reliable and promising alternative.

**Keywords:** stability analysis; berm; municipal solid waste (MSW) landfill; geotextile; factor of safety

## 1. Introduction

Geosynthetic-reinforced soil structures used as retaining berms in landfills have been reported by Qian and Koerner [1], Cowland [2], Sochovka [3], Espinoza and Houlihan [4], Eith et al. [5], and Gupta et al. [6] due to their advantages of simple design, convenient construction, low cost, and accommodation to large deformation; hence, they improve the overall stability of landfill slopes. The behavior of a geosynthetic-reinforced soil (GRS) berm in a landfill is complicated and full of uncertainties due to the variabilities of municipal solid waste properties (Eid et al. [7], Chen et al. [8], and Dixon and Jones [9]), the fluctuation of leachate levels (Jiang et al. [10] and Koerner and Soong [11]), and complex interactions between soil and reinforcements (Palmeira [12], Ferreira and Zornberg [13], Lee and Manjunath [14], and Lashkari and Jamali [15]). Therefore, the stability analysis of such berms is full of challenges.

Many studies on the stability of landfills have been carried out (Chen et al. [8], Kenter et al. [16], Koerner and Soon [17], and Stark et al. [18]), but only a few of them have studied the influence of berms on the stability of landfills (Qian and Koerner [1] and Zornberg and Kavazanjian [19]), and little research has been conducted to investigate the mechanical behavior and stability of berms themselves. Qian and Koerner [1] developed a three-part wedge method to analyze the translational failure of municipal solid waste (MSW) reinforced by soil berms that was designed with the aim of increasing

landfill space. However, the deformation of berms was not considered in their study, in which the berm was regarded as a rigid body and the slip surface needed to be assumed in advance to pass over the back slope of the berm or pass under the bottom of the berm. Zornberg and Kavazanjian [19] performed a finite element numerical simulation to assess the integrity of geogrid-reinforced steep slopes subjected to differential settlement and seismic loading because the finite element method can be used to monitor progressive failure up to and including overall shear failure, without assumption for the locations of slip surfaces. They found that the maximum predicted geogrid strains were indicated to well below the geogrid's allowable strain values; the failure mechanism was not investigated.

In this study, stability analyses were conducted to provide a better understanding for the behavior of the Xingfeng MSW landfill expansion accompanied by a GRS berm extension. The stability nalyses were performed using strength reduction method combined with FEM to obtain tensile force of geotextile, slip surface and factor of safety required by Chinese MSW landfill design standards (CJJ/176-2012 [20]) to be not less than 1.35 under normal working condition. The influence on the stability of geotextile strength, leachate level, and anti-slide pile arrangement were considered, with the aim to provide a quantitative safety evaluation and guidance for the application of the technique for GRS berm extension.

## 2. Project Background

The Xingfeng MSW landfill covers an area of 917,000 square meters, with a total storage capacity of 34.5 million tons. In terms of daily disposal capacity, the Xingfeng MSW landfill ranks second in China, only after the Laogang landfill in Shanghai. Since it was put into use in 2002, the speed of garbage disposal has been very fast, and the daily waste treatment amount has increased from 2381 tons in 2002 to 10,738 tons in 2017. In order to increase the storage capacity of the landfill, the berm on the south side needs be raised and expanded. Construction started in early 2016, and the berm body filling was basically completed in January 2017.

The original berm (or old berm) is a homogeneous clay impervious berm with a length of 300 m and a height of 20 m. The slope of the landfill behind the berm body is 1V:3H, and the height of the MSW mass is 65 m. A liner system is set between the MSW mass body and the berm to prevent leachate from entering the berm body. The berm is close to the leachate treatment plant (see Figure 1). As the site space is very limited, a steep slope for the berm has to be considered in the design; this is beyond the traditional ranges for a berm based on design guidelines. Finally, the berm was set with a gradient of 42.3° (1V:1.1H), reinforced by geotextiles, and laid in the whole section of the new berm with a vertical spacing of 0.5 m. A typical design section is shown in Figure 2. The height of the new reinforced berm is 35 m and the width of the berm crest is 12 m. A 2 m wide platform is set at the heights of 5, 15, and 25 m. A row of cast-in-place piles is used as anti-slide piles at the foot of the downstream slope, below which there is 15 m granite residual soil. A grass surface on downstream slope was adopted for the purpose of erosion control and aesthetic appearance.

The fill material was graded crushed stone, the particle size distribution of which is as follows: 20% of less than 5 mm, 20% of 5–10 mm, 20% of 10–20 mm, and 40% of 20–40 mm. The fill was laid and rolled layer by layer. The thickness of each layer of soil after compaction is not greater than 250 mm, and heavy machinery was used for compaction. The fill was compacted at a relative density of 95% with a unit weight 19.7 kN/m$^3$, which satisfies the field compaction requirement for GRS berms.

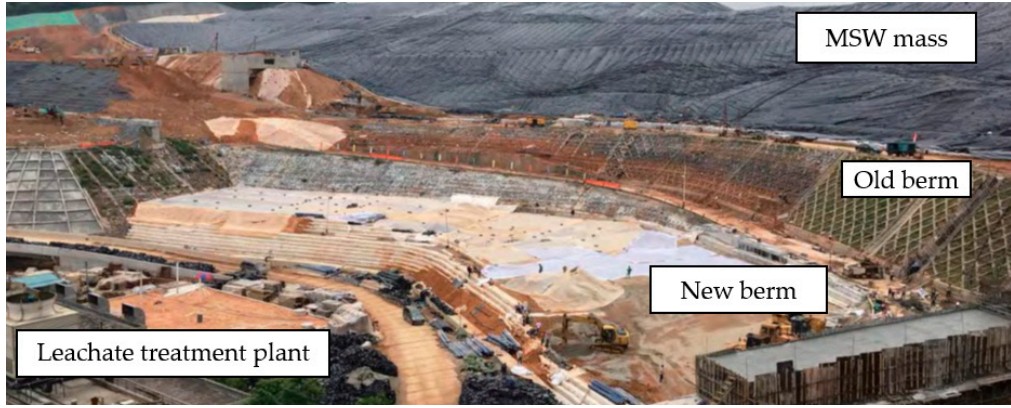

**Figure 1.** Photo of a new geosynthetic-reinforced soil (GRS) berm being raised and expanded.

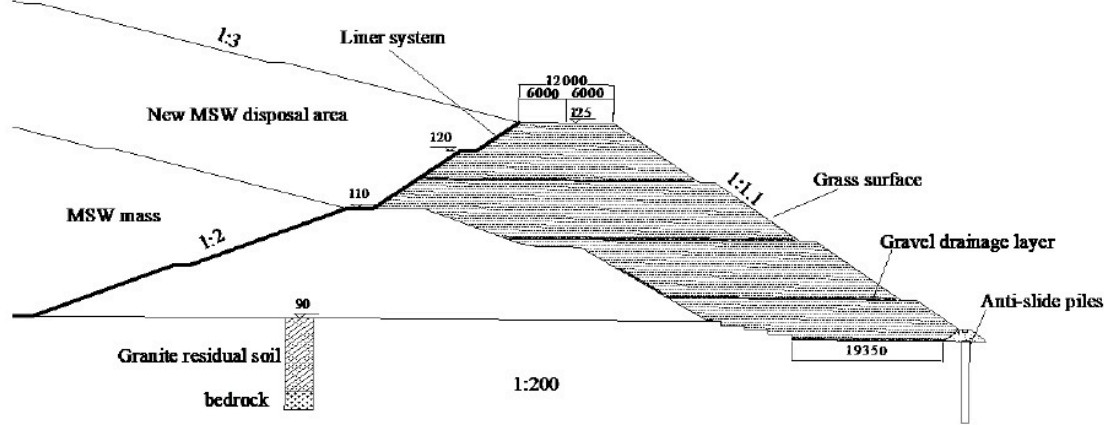

**Figure 2.** Typical cross section of the GRS berm (length: mm; height: m).

## 3. Numerical Model

The Plaxis 2D (Brinkgreve et al. [21]) program was used to examine the stability of the GRS berm, as it has been used by several researchers to investigate the factor of safety (FOS) of the reinforced slope with the strength reduction method (SRM) technique (Mohamed et al. [22], Tschuchnigg et al. [23], and Yang et al. [24]). A 135 m high and 780 m long numerical model mesh was considered to model the landfill with the GRS berm in a plane strain analysis, as illustrated in Figure 3. The bottom and lateral boundaries of the model domain were taken long enough from the berm to avoid any boundary effects. The bottom boundary was fixed against movements in all directions, whereas the vertical boundaries were restrained in the horizontal direction. The geotextile was simulated by the geotextile element embedded in Plaxis. The leachate level was estimated from field observation and boring logs, and leachate levels at the left slope surface of berm ($H_w$ in Figure 3) were set to 20 m for normal working conditions. Leachate was prevented from permeating into the berm with a liner system constructed between the MSW mass body and the berm.

After adopting the medium density defined in Plaxis 2D (Brinkgreve et al. [21]) program, the total number of elements was 4965 and the number of nodes was 40,199. The 15-node triangle was adopted as the default element because it provided a fourth order interpolation for displacements, and the numerical integration involved twelve Gauss points (stress points). An Intel(R) Xeon (R) CPU E3-1240 V5 @ 3.50GHz (Intel, Santa Clara, CA, United States) was used to perform the analyses, and the typical calculation time for one simulation was 60~90 s.

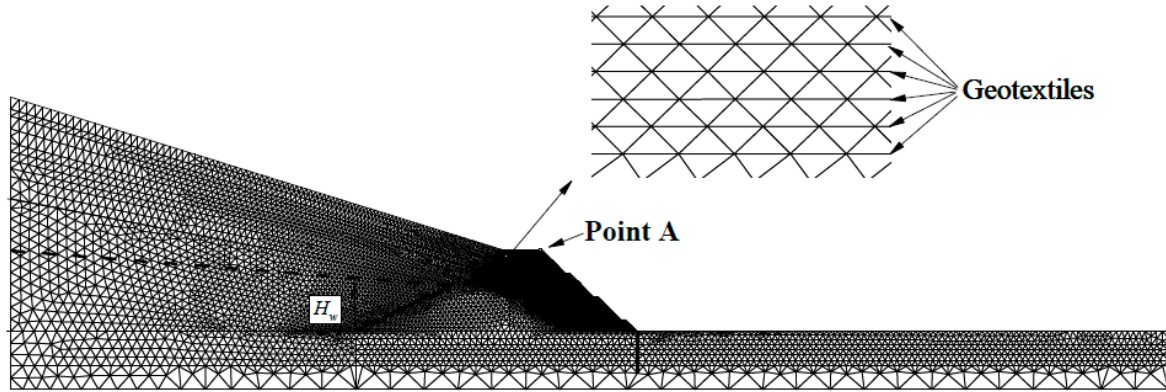

**Figure 3.** Finite element mesh.

### 3.1. Soil Propriety

To model the constitutive behavior of soil, an elastic perfectly plastic model based on Mohr–Coulomb failure criteria with a non-associated flow rule (Brinkgreve et al. [21]) was used with five parameters: cohesion (c), friction angle ($\varphi$), dilatancy angle ($\psi$), Young's modulus (E), and Poisson's ratio ($\upsilon$). Soil properties were obtained from the conducted laboratory tests, as shown in Table 1.

**Table 1.** Soil propriety of the Xingfeng municipal solid waste (MSW) landfill site. $\gamma$: unit weight; $w$: moisture content; $c$: cohesion; $\varphi$: friction angle; $\psi$: dilatancy angle; $E$: Young's modulus; and $\upsilon$: Poisson's ratio.

| Soil | $\gamma$ (kN/m$^3$) | $w$ (%) | $\varphi$ (°) | $\psi$ (°) | $c$ (kPa) | $\upsilon$ (-) | $E$ (MPa) |
|---|---|---|---|---|---|---|---|
| New berm | 19.7 | 11.2 | 32 | 0 | 10 | 0.3 | 50 |
| MSW | 10.8 | 35 | 15 | 0 | 12 | 0.35 | 10 |
| Old berm | 19 | 18 | 25 | 0 | 20 | 0.3 | 50 |
| Foundation | 20 | 24.8 | 23 | 0 | 33 | 0.25 | 63 |
| Bedrock | 25 | - | 42 | 0 | 300 | 0.25 | 5000 |

### 3.2. Geotextiles

A geotextile specimen was tested by wide-width tensile tests (ISO 10319 [25]) to determine long-term tensile strength. The test data are presented in load per unit width versus strain curve, from which the modulus values could be calculated.

$$J = \frac{T \times 100}{\varepsilon} \tag{1}$$

where $J$ is the secant tensile stiffness in kN/m, $T$ is the tensile force per unit width at a certain strain in kN/m, and $\varepsilon$ is the strain in %.

A strain of 5% was used because it was close to soil failure strain. The secant tensile stiffness and tensile force per unit width at 5% strain, $T_{5\%}$ and $J_{5\%}$, were 120 kN/m and 2400 kN/m, respectively, for the wide-width tensile tests (see Figure 4). The long-term tensile strength of geotextile at 5% strain was:

$$T_{5\%, l} = 120/6 = 20 \text{ kN/m} \tag{2}$$

The long-term secant stiffness at 5% strain was:

$$J_{5\%, l} = 2400/6 = 400 \text{ kN/m} \tag{3}$$

where the number of 6 is the reduction factor value that is typically in the range from 5 to 7, as suggested in the design guidelines (GB/T 50290-2014 [26]) accounting for the negative impact from creep, durability,

and installation damage. The reinforcement was modelled using a geogrid element built in Plaxis 2D with an axial stiffness and tensile strength on the basis of the tensile test results. The soil-reinforcement interface was assumed to be fully bounded in the numerical model because the primary failure mode of the GRS berm was not governed by the geotextile pullout failure (Hatami and Bathurst [27] and Yang et al. [23]).

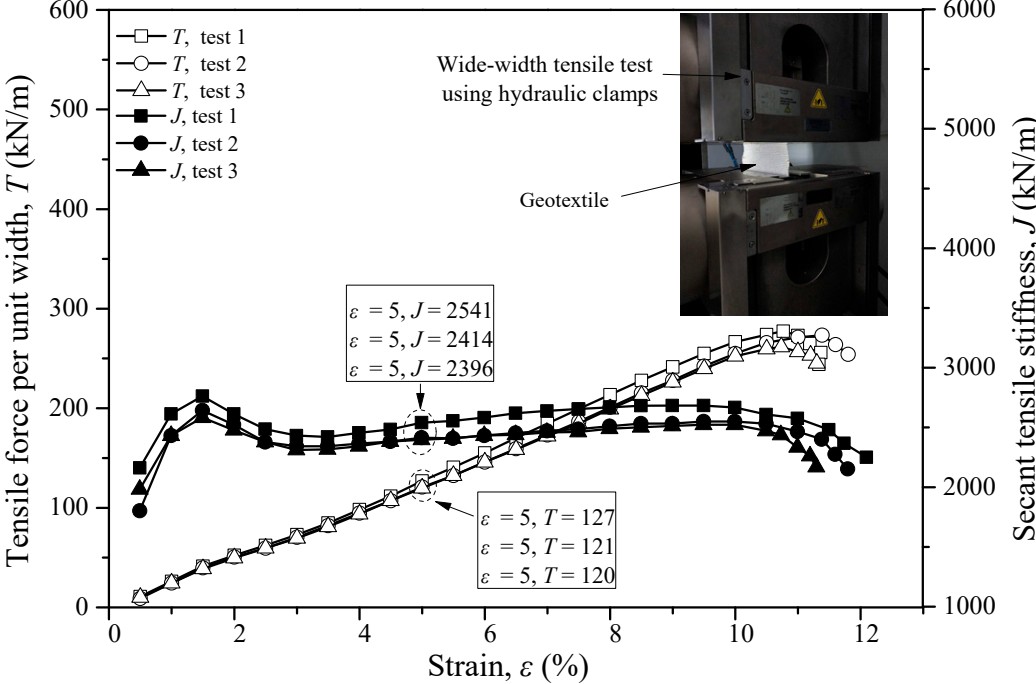

**Figure 4.** Load/unit width and secant tensile stiffness versus the strain curve of the geotextile.

### 3.3. Anti-Slide Piles

Anti-slide piles were selected to be 1.6 m in an equivalent diameter of the pile, along with a modulus of elasticity of 20,000 MPa and a Poisson's ratio of 0.2. The bending rigidity of the piles in the 2D model was $3.57 \times 106$ MN/m$^2$. The length of each pile in the model was 18 m. The piles' element was meshed using Mindlin beam element built in Plaxis 2D (Brinkgreve et al. [21]). The interface element between the soil layers and the pile elements was assumed to be fully bounded in the numerical model.

### 3.4. Strength Reduction Method

With the popularity of strength reduction method (Griffiths and Lane [28] and Dawson et al. [29]), many finite element (FE) method-based studies have been conducted to analyze the stability of GRS structures (Jie et al. [30], Tschuchnigg et al. [23], and Yang et al. [31])). The stability of the berm was calculated in terms of global factor of safety, which was evaluated using the SRM. In this approach, the FOS was taken as a factor by which the soil shear strength was reduced to bring the slope on the verge of failure. The factors of safety comprised the value of $\sum Msf$ at failure in the Plaxis program (Brinkgreve et al. [21]). The total multiplier $\sum Msf$ was used to define the value of the soil strength parameters at a given stage in the analysis:

$$\sum Msf = \frac{tan\varphi_{input}}{tan\varphi_{reduced}} = \frac{c_{input}}{c_{reduced}} = \frac{T_{5\%,l,input}}{T_{5\%,l,reduced}} = \frac{J_{5\%,l,input}}{J_{5\%,l,reduced}} \tag{4}$$

where the strength parameters with the subscript "input" refer to the properties entered in the material sets and parameters with the subscript "reduced" refer to the reduced values used in the analysis.

$J_{5\%,l}$ is long-term secant stiffness at a 5% strain of the geotextile and $T_{5\%,l}$ is long-term tensile strength at a 5% strain of the geotextile.

## 4. Numerical Model Validation

The FE model was verified by comparing the predicted results from the FE analyses with the measured displacement from centrifuge tests (unpublished report results) that were conducted in the Yangtze River research institute in Wuhan, China. The CKY-200 centrifuge of Yangtze River research institute was used in the experiment. Its effective capacity is 200 g-t, its maximum acceleration was 200 g, and its model box was $1.0 \times 0.4 \times 0.8$ m. In this experiment, a 100 g model was used, that is the scale ratio was 1:100. In order to conservatively evaluate the stability of the GRS berm in the centrifugal model test, an anti-slide pile was not considered, and the influence of anti-slide pile was taken as a safety reserve. The material of the berm body was taken from the coarse and fine particle mixed soil (excluding the soil particles with particle size greater than 30 mm). The maximum dry density of the filling material was 1.92 kg/m$^3$, and the optimal moisture content was 15.7 ($\pm$2) %. The drainage direct shear strength was 29.5 kPa and the friction angle was 36.4°. The MSW mass was simulated using loose fine sand, which only simulated the earth pressure of the MSW on the berm and ignored the simulation of the mechanical properties of the MSW itself; the thrust generated by the fine sand was slightly higher than that of the actual MSW so as to ensure that the centrifuge test was a conservative assessment of the stability of the refuse retaining dam. The average density of loose fine sand was 1.72 kg/m$^3$. Through narrow strip tensile and direct shear tests of the reinforcement soil interface, it was determined that a 0.45 mm thick polyester fiber was used to simulate the 2 mm thick geotextile, according to the similar friction and equivalent tensile strength per unit area between the prototype and the centrifuge model. The berm soil was layered by 0.05 m and compacted to a density of $\geq$1.85 kg/m$^3$ by manual hammering at the optimum moisture content. Polyester was laid in a 0.05 m layer, and the reverse wrapping was carried out on the slope surface to simulate the reverse wrapping process of the geotextile in practical engineering.

Five laser displacement monitoring points were set up on the slope to measure its horizontal and vertical displacement. Figure 5 shows a comparison results of lateral displacement and settlement between measured values from test and predicted from the FE analysis. The measured lateral displacement and settlement of the monitored point in the centrifuge mode were approximated to the values predicted in the FE analysis. As can be seen in Figure 5, the FE and centrifuge models showed good agreement in the displacement field, thus demonstrating the capability of FE modelling to predict the behavior of a GRS berm.

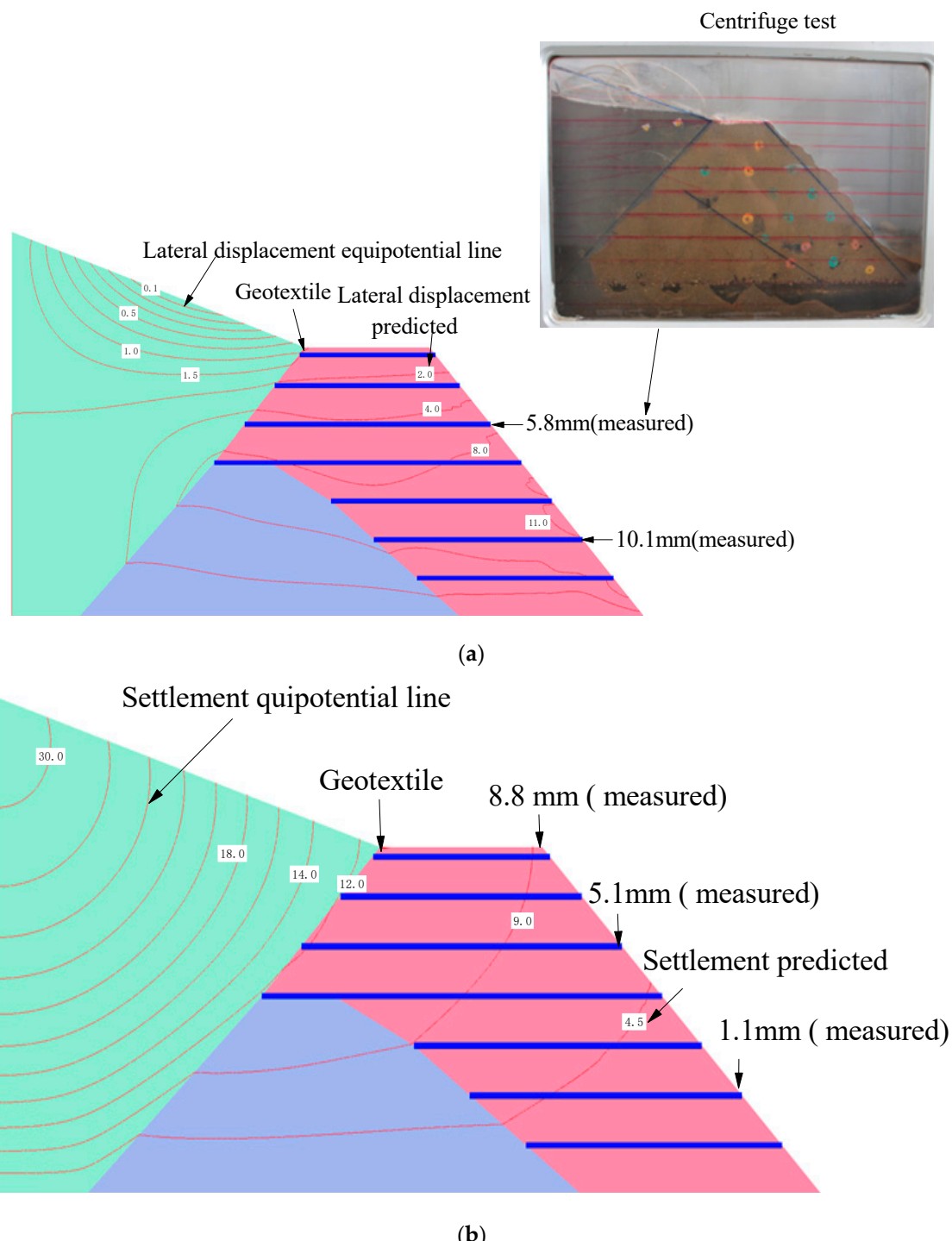

**Figure 5.** Comparison of berm's displacements between the numerical results and the centrifuge test data: (**a**) Lateral displacement (mm) and (**b**) settlement (mm).

## 5. Results and Discussion

The development of $\sum Msf$ was plotted as a function of the displacement of the berm crest point A (see Figure 3), as shown in Figure 6. It is clear that as the displacement of point A reached a value of 0.6 m, the $\sum Msf$ gave a value of approximately 1.42, which meant that the FOS was 1.42—larger than the 1.35 required by specification (CJJ/176-2012 [20]). Figure 7 illustrates the failure mechanism of the GRS berm at normal working conditions by means of a nodal displacement vector. The sliding surface went through the toe of the berm, developed along the interface between the new and old

berms, and finally extended into the MSW mass on the upper left of the GRS berm. It seemed that the anti-slide piles effectively blocked the movement of foundation soils. The maximum tensile force of all geotextiles was 4.9 kN/m, far lower than the long-term tensile strength of the geotextile at a 5% strain $T_{5\%,l}$ = 20 kN/m, which suggested that the geotextile had enough of a safety reserve.

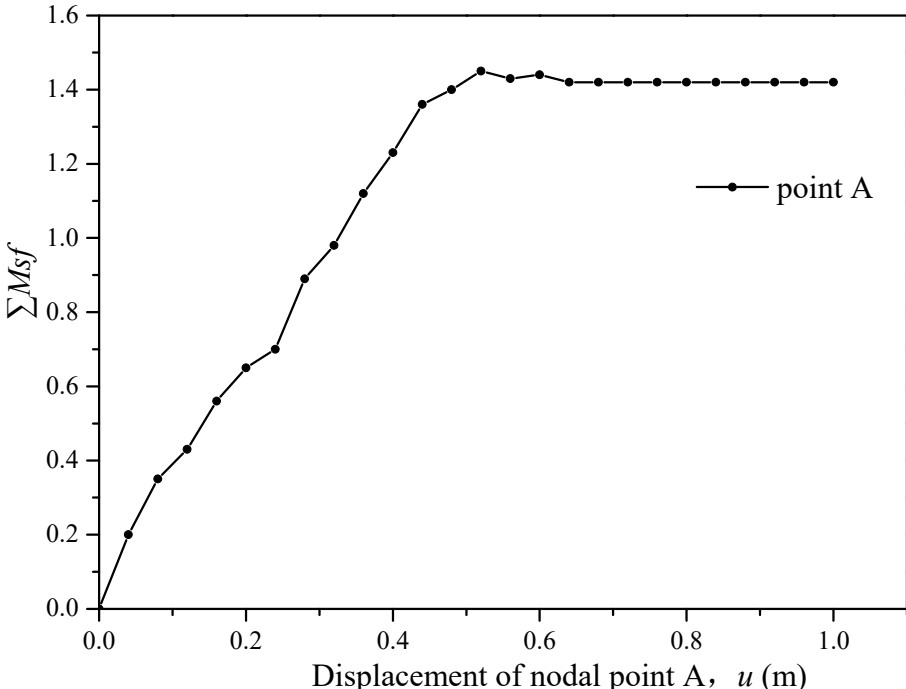

**Figure 6.** $\sum$Msf (equal to the factor of safety (FOS)) versus the displacement of nodal point A at the berm crest.

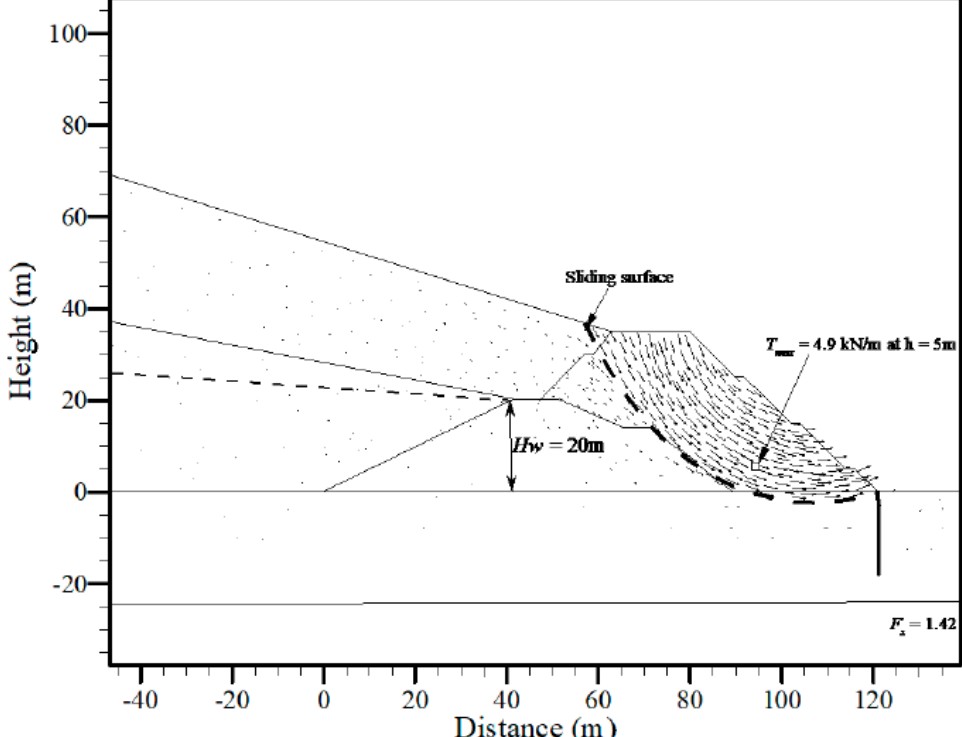

**Figure 7.** Numerical result of baseline under normal working conditions.

A series of parametric studies was conducted using a similar numerical model with slight changes to certain points, including the effect of several key parameters that were expected to control the stability of the GRS berm: (1) geotextile strength, (2) leachate level, and (3) anti-slide pile arrangement. Three group analyses are listed in Table 2.

**Table 2.** Parametric study performed on the GRS berm.

| Cases | | $S_v$ | $T_{5\%,l}$ | $L_{pile}$ | $H_w$ | Description |
|---|---|---|---|---|---|---|
| | | m | kN/m | m | m | |
| Baseline | | 0.5 | 20 | 18 | 20 | Design scheme |
| Group I | Geotextile I | 1.0 | 15 | 18 | 20 | To investigate effect of geotextile strength |
| | Geotextile II | 1.5 | 10 | 18 | 20 | |
| Group II | Leachate level I | 0.5 | 20 | 18 | 15 | To investigate effect of leachate level |
| | Leachate level II | 0.5 | 20 | 18 | 25 | |
| | Leachate level III | 0.5 | 20 | 18 | 30 | |
| Group III | Piles I | 0.5 | 20 | 10 | 20 | To investigate effect of pile arrangement |
| | Piles II | 0.5 | 20 | 25 | 20 | |
| | Piles III | 0.5 | 20 | 10 + 10 | 20 | |

### 5.1. Effect of Geotextile Strength

To explore the influence of the geotextile's strength on the behavior of the GRS berm, a group of analyses was carried out by varying the long-term tensile strength of the geotextiles at 5% strain to $T_{5\%,l} = 20$, 15, and 10 kN/m at a geotextile spacing of $S_v = 0.5$ m.

The failure mechanisms of the GRS berm with different geotextile strengths are plotted in Figure 8a–c. It is clear that the failure surface tended to extend to the MSW zone with the higher strength of the geotextile (i.e., $T_{5\%,l} = 20$ kN/m in Figure 7). With the decreasing geotextile strength, the failure surface became more confined within the GRS berm (Figure 8b,c).

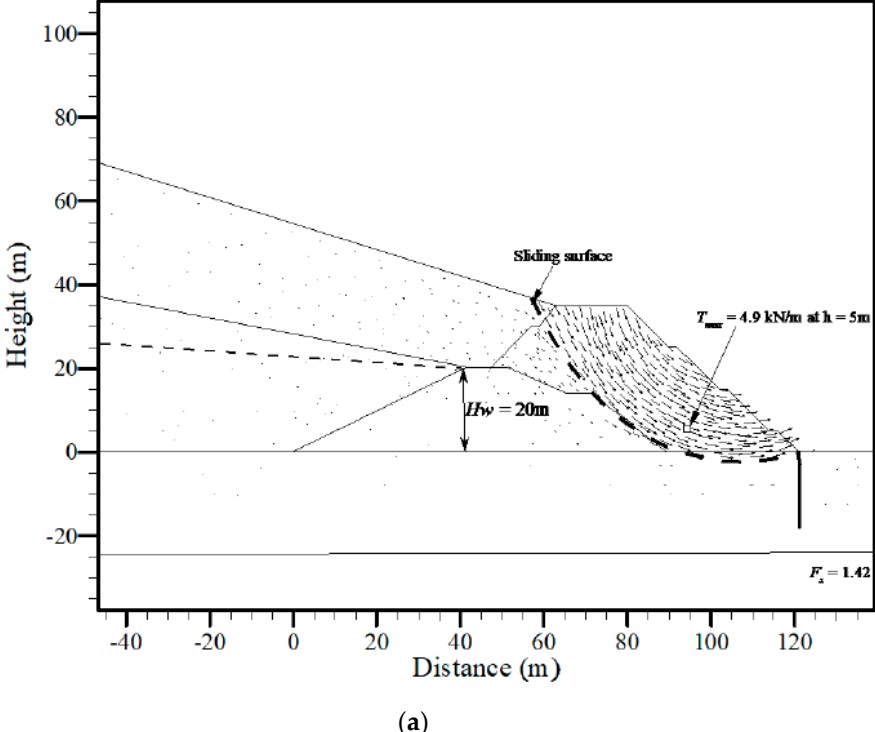

(**a**)

**Figure 8.** *Cont.*

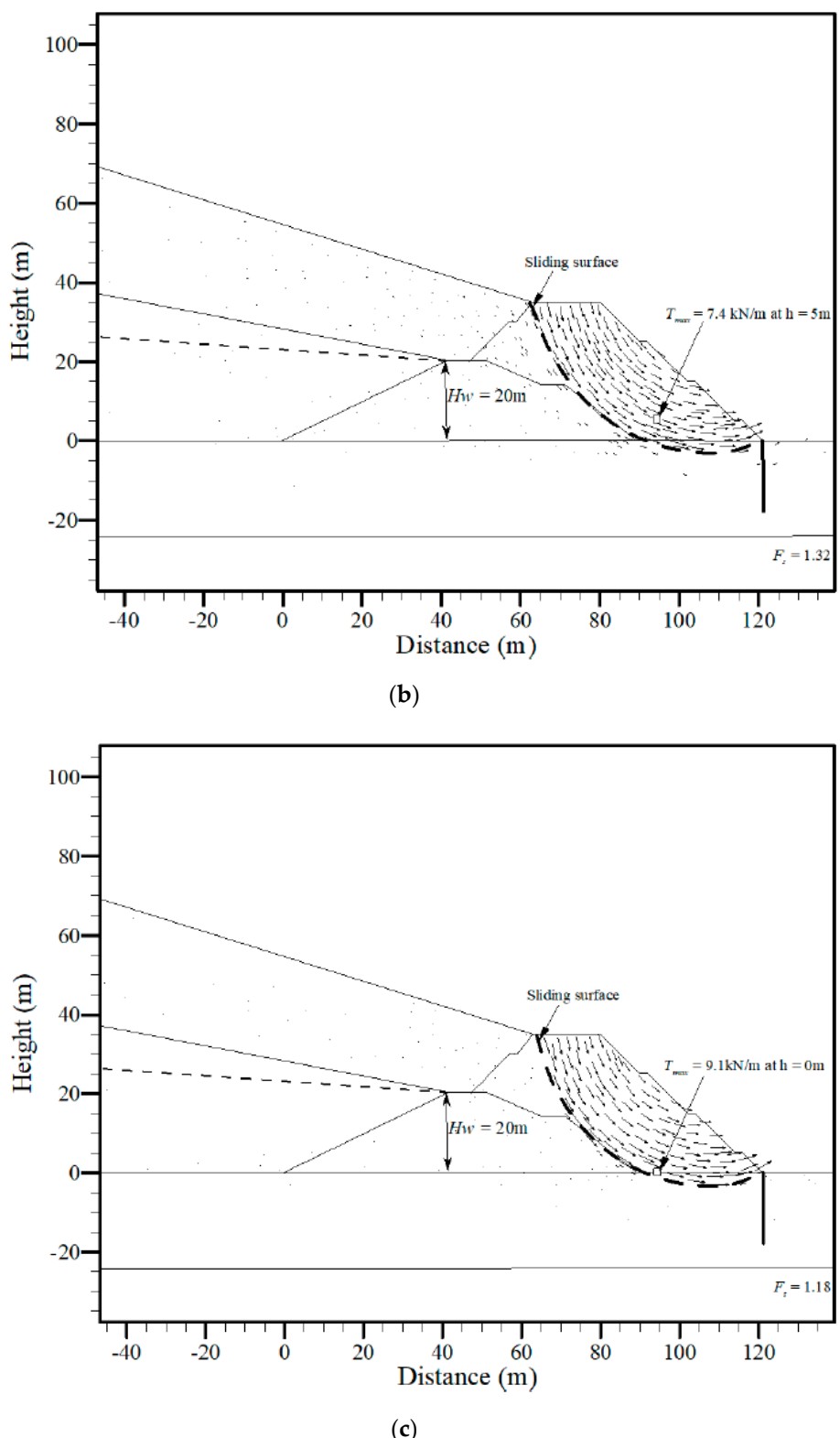

**Figure 8.** Effect of geotextile strength: (**a**) Baseline ($T_{5\%,l}$ = 20 kN/m), (**b**) Geotextile I ($T_{5\%,l}$ = 15 kN/m), and (**c**) Geotextile II ($T_{5\%,l}$ = 10 kN/m).

Figure 9 depicts the influence of geotextile strength on the factors of safety and the maximum tensile force of the geotextile reinforcement in the GRS berm. It was apparent that the factors of safety of the GRS berm decreased with the decreasing geotextile strength. Consequently, the lower geotextile

strength led to the increased maximum tensile force of the geotextile. Similar findings were found through centrifuge models in a series performed by Zornberg et al. [32] to investigate the effect of geotextile tensile strength on the stability of reinforced slopes. Failure surfaces that developed in the models in this series showed that models built with stronger geotextiles as reinforcement elements did not exhibit the sudden collapse observed in models built using the weaker fabric. This means that a higher strength geotextile improves slope stability.

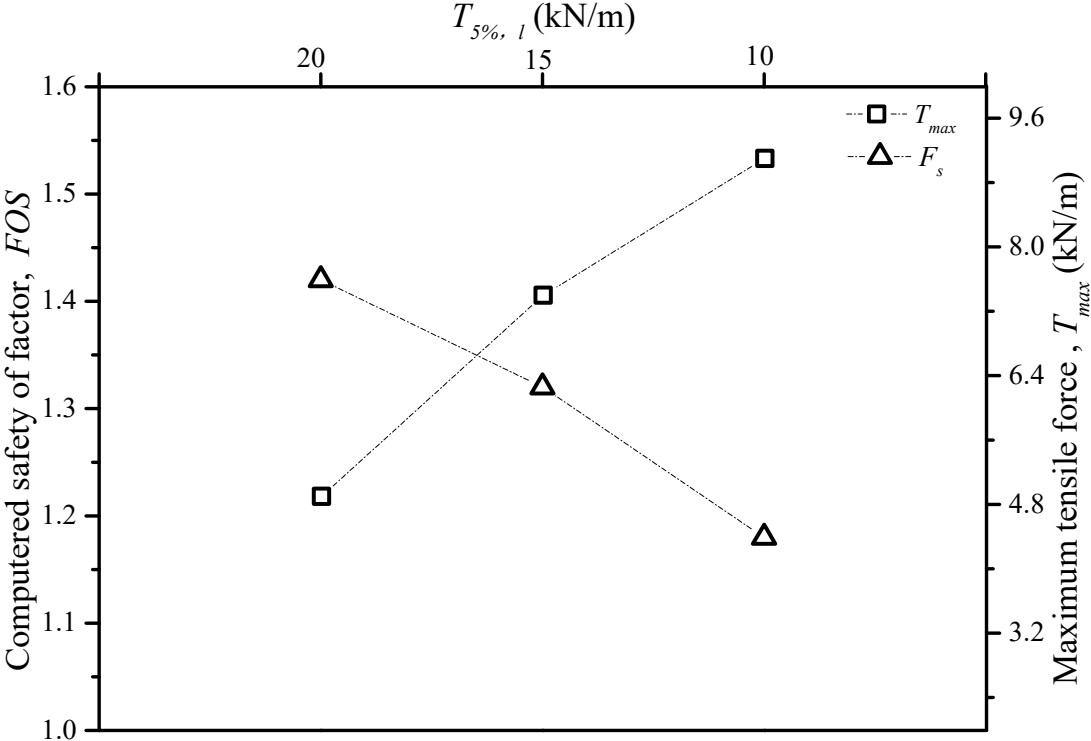

**Figure 9.** FOS and $T_{max}$ versus geotextile strength.

*5.2. Effect of Leachate Level*

To illustrate the effect of the leachate level on the failure mechanism, the failure surfaces and corresponding factors of safety are displayed in Figure 10 with various leachate levels of $H_w$ = 15, 20, 25, and 30 m. Other parameters were kept constant. Through a comparison with Figure 10a–d, it can be found that all the failure surfaces developed along the interface between the new and existing berm. The leachate level had a significant impact on the factor of safety of the berm—the higher of the leachate level was, the lower the factor of safety.

The effect of the leachate level on the stability and the maximum tensile force of the geotextiles is displayed in Figure 11. It is apparent that with the increasing leachate level, the factor of safety decreased and the maximum tensile force of the geotextiles increased. Once the factor of safety approached FOS = 1.0, the tensile force of the geotextile dramatically increased. A significant effect of the leachate level was also observed in slope stability studies by Jiang et al. [10], Koerner and Soong [11]. Jiang et al. [10] indicated that when there was no leachate in the landfill, the FOS levels were 2.22 and 2.14 for landfill units I and II, respectively. When the leachate level saturated, the FOS for units I and II were 1.14 and 0.15, respectively. Koerner and Soong [11] illustrated a case history where the leachate level was known to be high, i.e., 16–30 m above the liner, and they showed how the FOS decreased as the leachate level increased. Clearly, cases of high leachate levels on liner systems can lead to stability failure, as illustrated in the two cases.

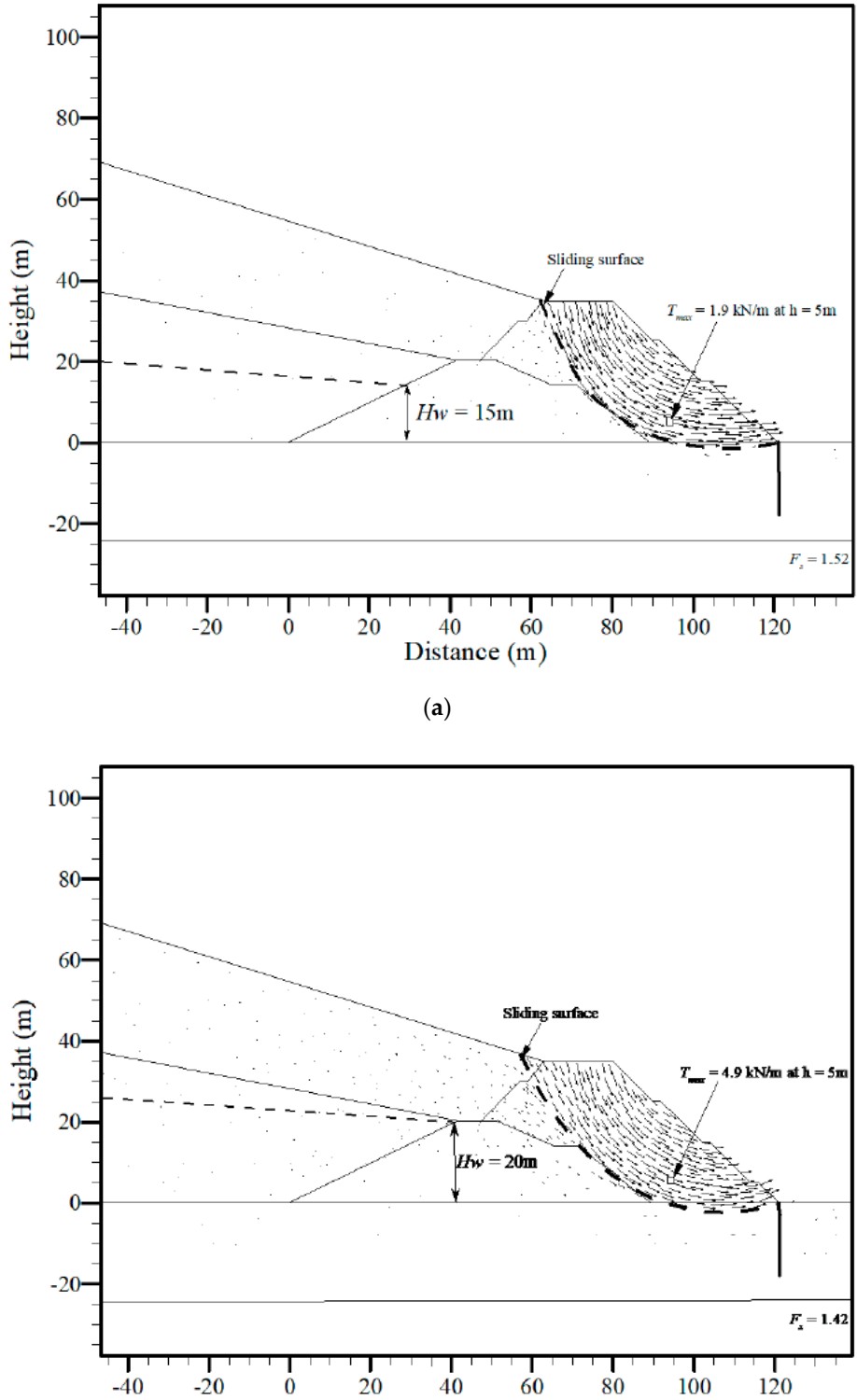

(**a**)

(**b**)

**Figure 10.** *Cont.*

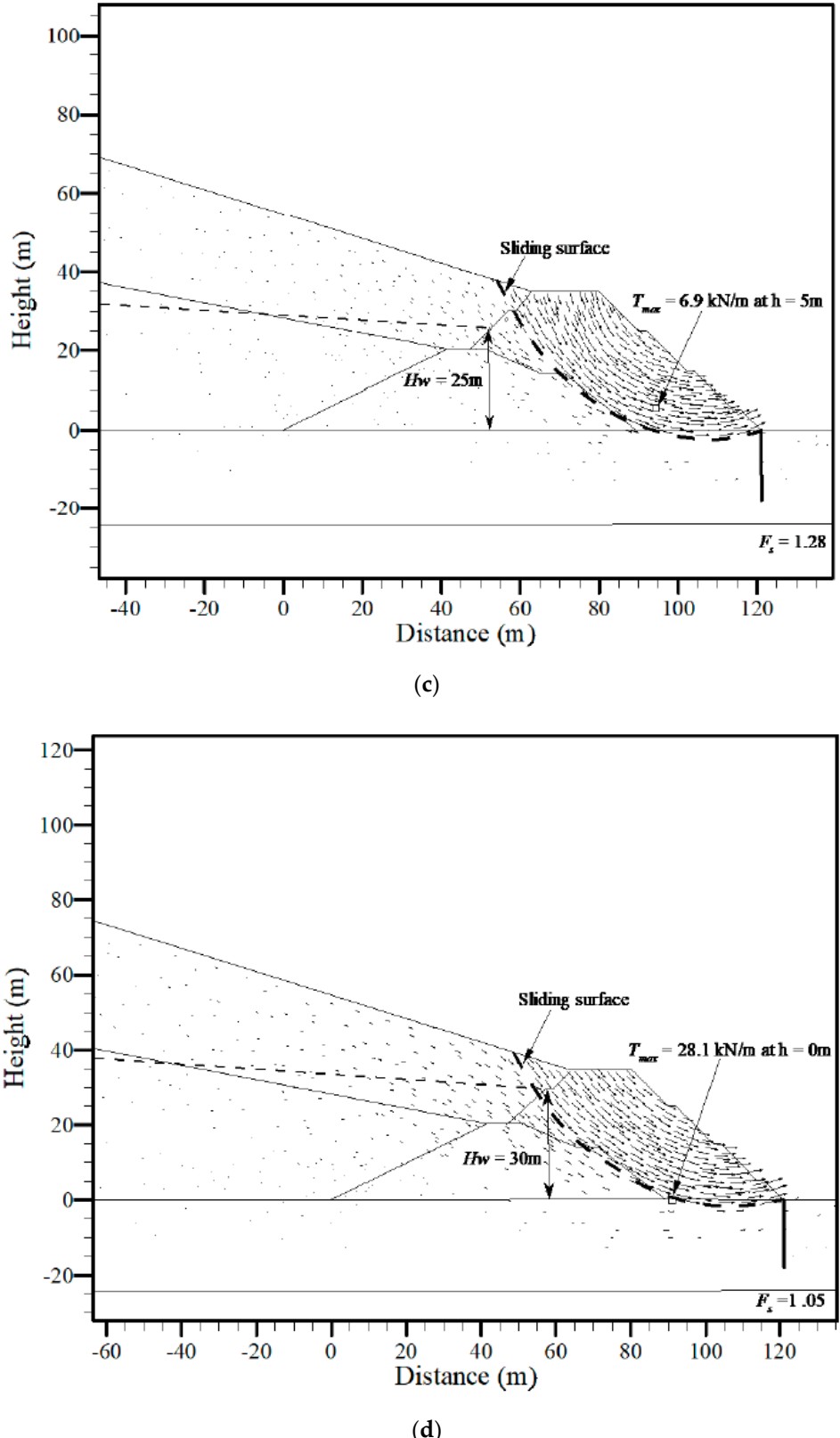

**Figure 10.** Effect of leachate level: (**a**) Leachate level I ($H_w$ = 15 m), (**b**) Baseline ($H_w$ = 20 m), (**c**) Leachate level II ($H_w$ = 25 m), and (**d**) Leachate level III ($H_w$ = 30 m).

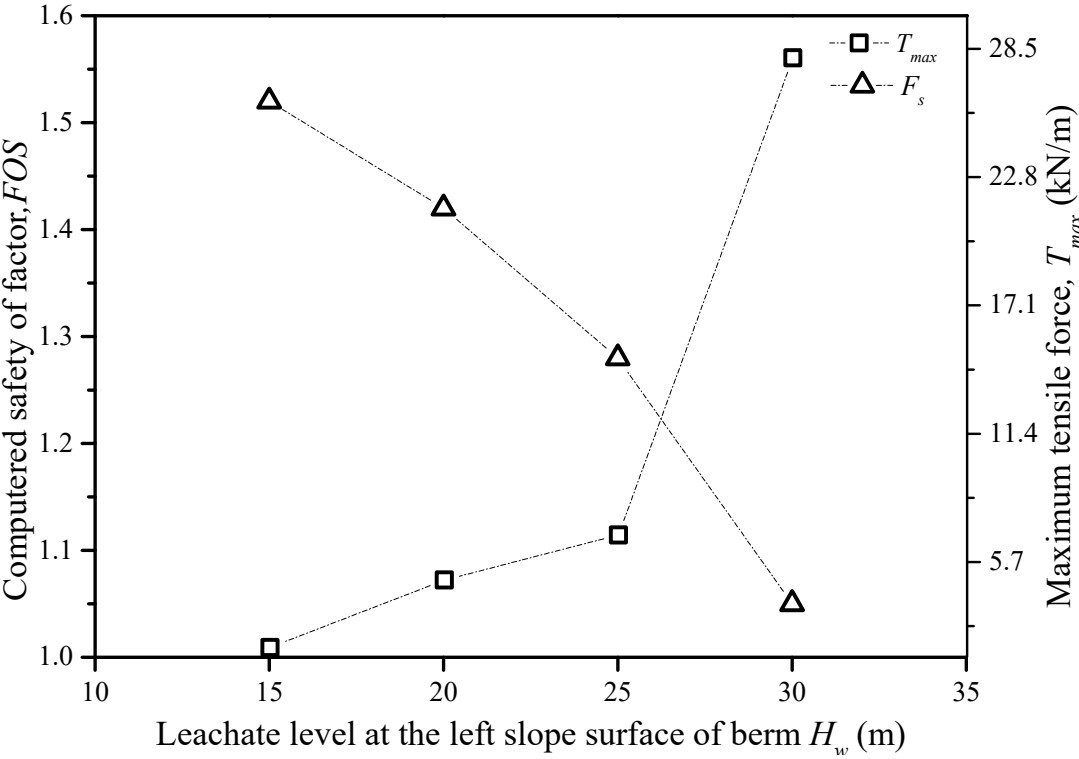

**Figure 11.** FOS and $T_{max}$ versus leachate level at the left slope surface of berm $H_w$.

### 5.3. Effect of Pile Arrangement

To check the effect of pile arrangement on the behavior of the GRS berm, four cases with different pile arrangement were analyzed (Baseline and Group III in Table 2). The four cases included: Case I—one row of piles 10 m in length (Group III—Piles I in Table 2); Case II—one row of piles 18 m in length (Baseline in Table 2); Case III—two rows of piles 10 m in length and 5 m in spacing (Group III—Piles II in Table 2); and Case IV—one row of piles 25 m in length (Group III—Piles III in Table 2). The failure mechanisms of these four cases are shown in Figure 12a–d, and the pile arrangement effect on the factor of safety and the maximum tensile force of the geotextile is shown in Figure 13. Through comparison, it could be seen that doubling the row of piles (hence doubling the total pile length) had a minimal effect on the factor of safety ($F_s$ = 1.20 and 1.25). However, by comparing Cases I, II, and IV with only one row of piles but increasing pile length (i.e., $L_{pile}$ = 10, 18, and 25 m, respectively), the factor of safety increased quite dramatically. By comparing Cases II and III with a similar total pile length (i.e., 18 and 20, respectively), the two row of piles showed a lower efficiency in increasing the factor of safety of the berm. As all factors of safety of these four cases were $F_s$ = 1.20 or higher, the effect of the pile arrangement on the maximum tensile force of the geotextile was minimal. Zornberg and Kavazanjian [19] reported a case history in which concrete piers were used as a deep foundation to enhance the stability of a geogrid-reinforced steep landfill slope at the Operating industries, Inc. (OII) Superfund site, a hazardous waste site in southern California. However, the effect of pile arrangement on the GRS structure was ignored.

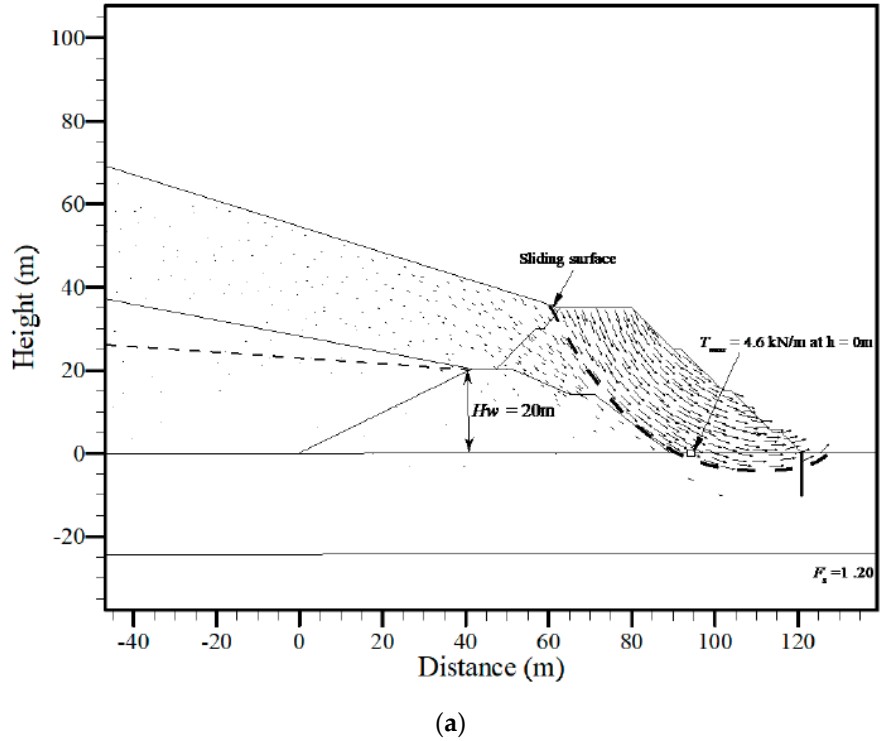

(**a**)

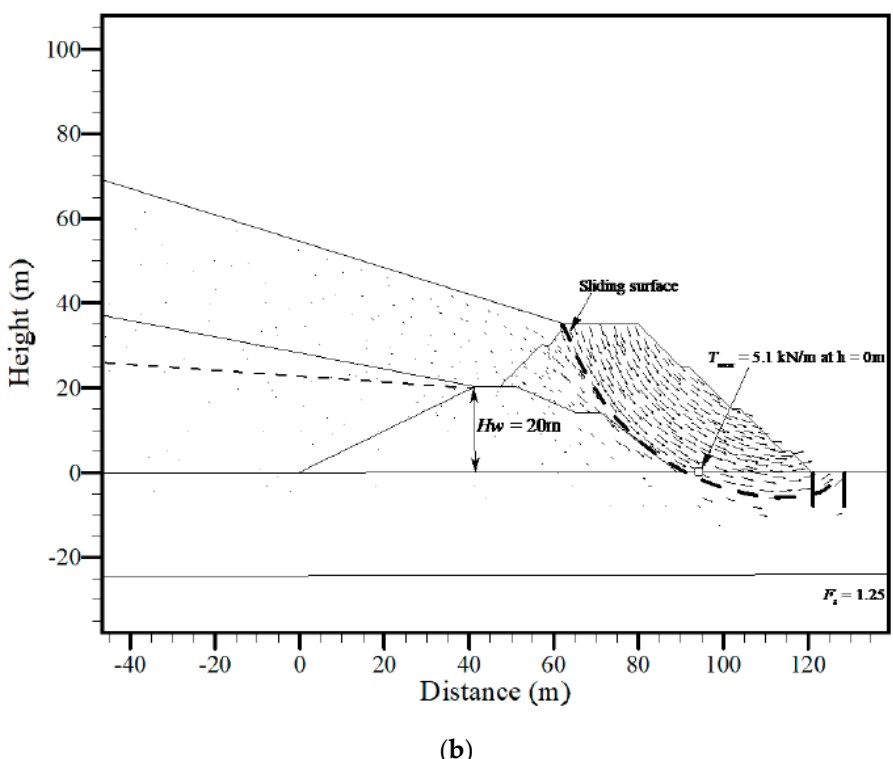

(**b**)

**Figure 12.** *Cont.*

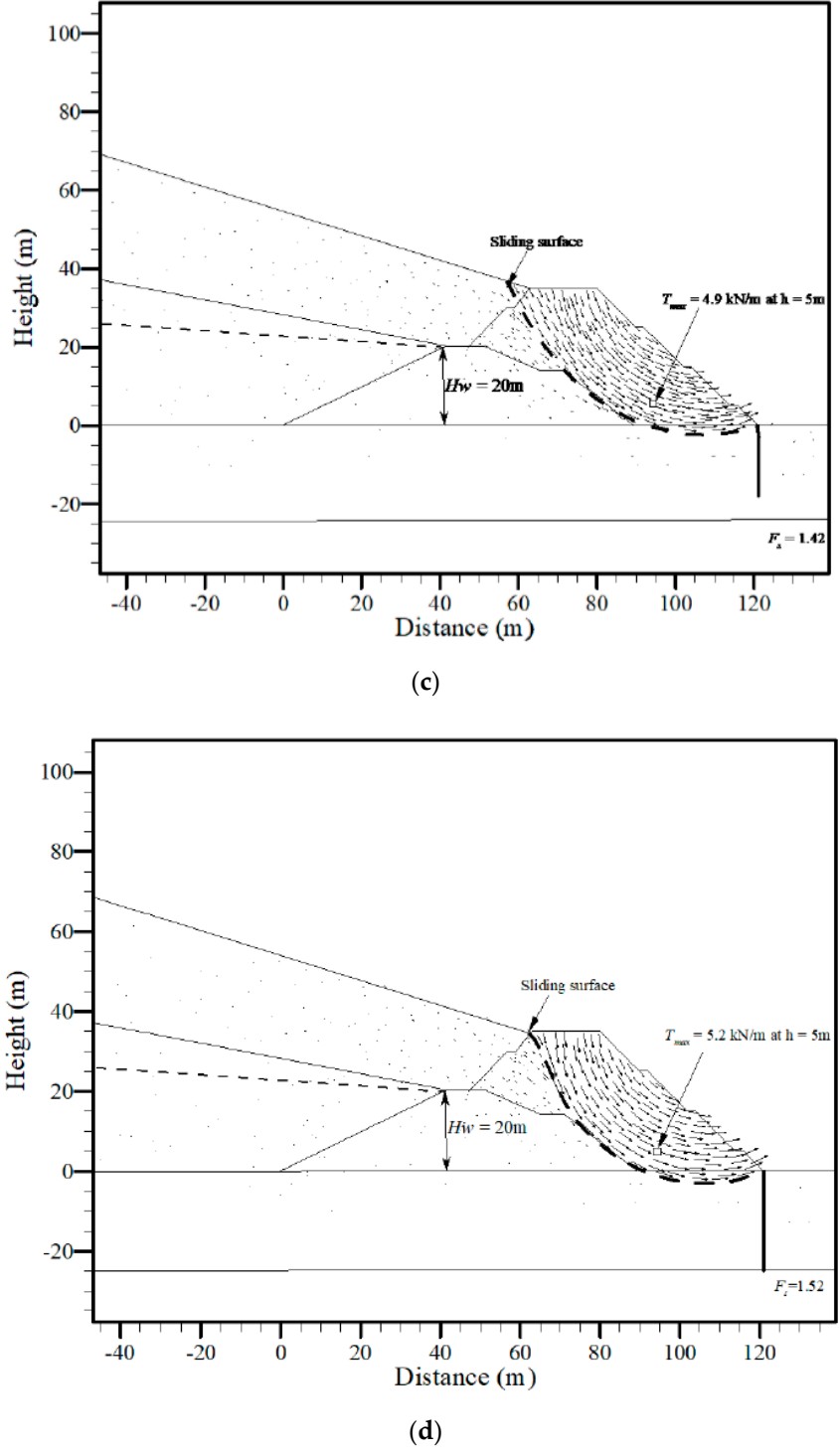

**Figure 12.** Effect of pile arrangement: (**a**) Piles I ($L_{pile}$ = 10 m), (**b**) Piles II (double $L_{pile}$ = 10 m), (**c**) Baseline ($L_{pile}$ = 18 m), and (**d**) Piles III ($L_{pile}$ = 25 m).

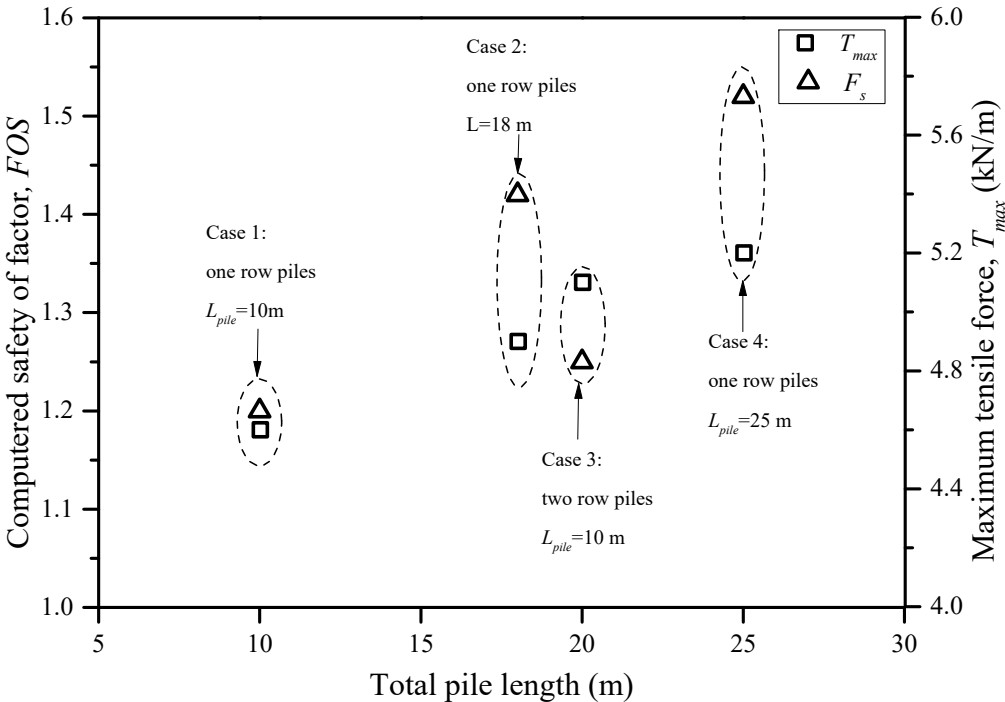

**Figure 13.** FOS and $T_{max}$ versus total pile length (m).

## 6. Conclusions

This paper presents a numerical study of a GRS berm extension project in the MSW landfill with a height of 35 m and a steep slope with a gradient of 42.3° (1V:1.1H). A series of FE analyses were performed to study the stability and effects of influencing factors including geotextile strength, leachate level, and anti-slide pile arrangement. It was found that:

(1) The leachate level has significant impact on the stability of a berm. The operational leachate level should be controlled at a height of 20 m; in such a case, an FOS value of 1.42 can be obtained and thus meet the specification (CJJ/176-2012) requirement of 1.35.

(2) The stability of the berm is higher when it is reinforced with a higher strength geotextile and the predicted tensile force of the geotextile is lower.

(3) An anti-slide pile arrangement has a certain influence on the slip surface. The failure surface goes through the toe of the berm, develops along the interface between the new and old berms, and finally extends into the MSW mass on the upper left of the GRS berm.

(4) When the total length of piles is the same, the reinforcement effect of single row of piles is better than that of a double row of piles.

The construction of this project described above was completed in 2017 and has been put into use for more than three years without any safety problems. Therefore, the design and analysis of this project can be used as reference for landfill expansion—especially for a site condition with limited space, a GRS berm is a safe, reliable, and promising alternative.

**Author Contributions:** Data curation, H.J.; formal analysis, H.J. and X.Z.; funding acquisition, X.Z.; methodology, H.J. and X.Z.; software, H.J. and Z.X.; writing—original draft, H.J., X.Z., and Z.X. All authors have read and agreed to the published version of the manuscript.

**Funding:** The research was funded by Science and technology program of Guangzhou (201707020047). Grateful appreciation is expressed for the support.

**Acknowledgments:** We sincerely acknowledge the former researchers for their excellent work, which greatly assisted our academic study.

**Conflicts of Interest:** The authors declare no conflict of interest.

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
