# Peer review of "Stability of Extended Earth Berm for High Landfill"

_applsci, doi:10.3390/app10186281_

Round 1

Reviewer 1 Report

General Comments:

Using Plaxis 2D (finite element analysis), the authors have numerically analyzed the stability of a berm located in China while focusing on leachate level, tensile strength of geotextiles, anti-slide pile arrangement, and length of piles. The manuscript provides some useful outcomes as discussed in the results and outlined in the conclusions section regarding the application of geotextile-reinforced extended berms. However, the authors do not compare their findings with previous literature. For example, the leachate level has an impact on the stability of a berm – this is a known characteristic of berms. The authors need to compare their findings with other literature. The rest of the conclusions are also quite generic in terms of research findings. The authors need to come up with a discussion section explaining why their research is important and how their findings can be utilized. They need to answer a set of questions. For example, is the current construction agrees with their findings? If not, what do the authors recommend?

In general, the manuscript is brief and to the point. The figures are well described and the results are reported in a reader-friendly manner. Once the authors compare and contrast their findings with relevant literature dealing with similar features of geotextile-reinforced berms, I would recommend publication. At this point, I recommend a revision.

The entire manuscript also requires an editorial review in terms of the English language. There are grammatical and punctuation errors throughout the manuscript. The citing style is very unusual and hinders readability. Several typos need to be fixed. I recommend proofreading the manuscript by a native English speaker.

I have a few specific comments as well.

Specific Comments:

Line 25: Please include the full form of GRS (Geosynthetic-Reinforced Soil).

Line 81: What are FOS and SRM? Please add a description or definitions of these terms.

Line 143: Please abbreviate “Finite Element” to FE before using it.

Line 156: There is no “point A” in Figure 3. At least it’s not visible.

Table 2: Please add some description to the outcomes of this table. What am I supposed to learn from this table?

Line 176: trends to extent >> tends to extend

Line 186: Why Holtz is cited as “0”? It should be 32.

Lines 193-195: The statements are not clear. Please rewrite.

Line 241: “By comparing t n be seen….” >> Looks like some words are missing here.

Author Response

Thank you very much for your valuable comments. Please refer to the attachment for detailed modification.

Reviewer 2 Report

  • Rephrase sentence at lines 15-18
  • Define all the abbreviations used in the paper. Do this either at the beginning of the article or, preferably, every time a new abbreviation is introduced in the text (please refer to the journal specifications). Some of the abbreviations that are missing are: GRS, MSW, FOS, SRM, etc.
  • Rephrase sentence at lines 45-48
  • Indicate the number of DOFs used in the numerical model, and the typical calculation time for one simulation, plus information about CPU used to perform the analyses
  • The information about the centrifuge test is too poor: more information is needed to be able to convince the reader that your model is validated.
  • Add a justification of the model parameters: were they derived/assumed or were they calibrated based on the results of the centrifuge tests? if the latter strategy was used, this has to stated when introducing the model parameters and the title of section 4 'Model Validation' has to be changed
  • Figure 5: it is difficult to see the results of the centrifuge tests, please provide a better figure, or an additional figure for these tests.

Author Response

(The authors gave the same response as above.)
